

Manuscript prepared for Atmos. Chem. Phys.
with version 2015/04/24 7.83 Copernicus papers of the LaTeX class copernicus.cls.
Date: 8 July 2016

# Dynamic sub-grid heterogeneity of convective cloud in a global model: Description and Evaluation of the Convective Cloud Field Model (CCFM) in ECHAM6–HAM2

Zak Kipling[1], Philip Stier[1], Laurent Labbouz[1], and Till Wagner[1,*]

[1]Department of Physics, University of Oxford, Oxford, UK
[*]now at Guy Carpenter & Company GmbH, Magnusstr. 11, 50672 Cologne, Germany

*Correspondence to:* Zak Kipling (zak.kipling@physics.ox.ac.uk)

**Abstract.** The Convective Cloud Field Model (CCFM) attempts to address some of the shortcomings of both the commonly-used bulk mass-flux parameterisations, and those using a prescribed spectrum of clouds. By considering the cloud spectrum as a competitive system where cloud types interact through their environment in competition for convective available potential energy (CAPE),

the spectrum is able to respond dynamically to changes in the environment. An explicit Lagrangian entraining plume model for each cloud type allows the representation of convective cloud microphysics, paving the way for the study of aerosol–convection interactions at the global scale where their impact remains highly uncertain.

In this paper, we introduce a new treatment of convective triggering, extending the entraining

plume model below cloud base to explicitly represent the unsaturated thermals which initiate convection. This allows for a realistic vertical velocity to develop at cloud base, so that the cloud microphysics can begin with physically-based activation of cloud condensation nuclei (CCN). We evaluate this new version of CCFM in the context of the global model ECHAM6–HAM, comparing its performance to the standard Tiedtke–Nordeng parameterisation used in that model.

We find that the spatiotemporal distribution of precipitation is improved, both against a climatology from the Global Precipitation Climatology Project (GPCP) and also against diurnal cycles from the Tropical Rainfall Measurement Mission (TRMM) with a reduced tendency for precipitation to peak too early in the afternoon. Cloud cover is quite sensitive to the vertical level from which the dry convection is initiated, but when this is chosen appropriately the cloud cover compares well with

that from Tiedtke–Nordeng.

CCFM can thus perform as well as, or better than, the standard scheme while providing additional capabilities to represent convective cloud microphysics and dynamic cloud morphology at the global scale.



## 1 Introduction

Clouds play a major role in the climate system, in terms of the radiation budget, the hydrological cycle and atmospheric dynamics. Their effects remain some of the largest uncertainties in estimates of climate sensitivity and current and future anthropogenic forcing (Boucher et al., 2013; Myhre et al., 2013).

Cloud parameterisations in global models typically have a sharp divide between large-scale stratiform clouds which can be resolved on the model grid, and sub-grid-scale convective clouds which cannot. While it is common for large-scale cloud and precipitation schemes to include detailed microphysics and prognostic condensate, cloud fraction and hydrometeor size distributions, with an explicit link to aerosol via droplet activation, the representation of in-cloud processes in convective clouds is generally much more simplistic.

Most current global atmospheric general circulation models (AGCMs) use one of a variety of bulk mass flux parameterisations for convection (e.g. Tiedtke, 1989; Kain and Fritsch, 1990; Bechtold et al., 2001). With a suitable closure, these provide a computationally efficient way of representing convective clouds in terms of the total updraught and downdraught mass fluxes in a grid column given the resolved-scale thermodynamic profile. However, neither the vertical velocity nor the horizontal area of these updraughts and downdraughts is represented; nor is the heterogeneous nature of convective clouds at the grid scale. This makes the representation of aerosol activation, ice nucleation and size-resolved microphysics problematic, although there have been limited attempts to include them in parameterisations of this type. However, these are precisely the processes through which atmospheric aerosol may exert many of its effects on the development of convective clouds (Lohmann and Feichter, 2005; Rosenfeld et al., 2008).

There are alternatives to the bulk mass flux approach, however. In superparameterisation (Grabowski, 2001; Khairoutdinov and Randall, 2001), a cloud-resolving model (CRM, typically 2D) is coupled to each column of the AGCM. This is an effective approach allowing for explicit representation of many aspects of convective cloud, but currently too computationally expensive for long climate simulations. Donner (1993) and Donner et al. (2001) emphasise cloud and mesoscale structures rather than mass fluxes, allowing cloud-system development and microphysics to be represented more precisely, but the semi-empirical nature of certain aspects may limit the generality of these schemes.

As another alternative to the bulk mass flux approach, spectral parameterisations have also been around for several decades, mostly based on Arakawa and Schubert (1974). Rather than a homogeneous field of average convective updraughts, these represent a range of different updraught/cloud types each with its own properties, typically defined by their fractional entrainment rates. While most parameterisations of this type prescribe the cloud spectrum empirically, the Convective Cloud Field Model (CCFM; Nober and Graf, 2005; Wagner and Graf, 2010) predicts the spectrum based on the competitive interactions between different cloud types. Coupled with an explicit entraining



plume model for each cloud type, this provides a promising setup in which to investigate the effects of convective microphysics at the global scale.

So far, CCFM has been evaluated in a single-column model (Wagner and Graf, 2010) and an earlier version was evaluated in a regional model (Graf and Yang, 2007). In this paper, we describe CCFM as it is currently implemented as an extension to the ECHAM–HAMMOZ global model,

including the addition of a sub-cloud dry convection treatment for triggering and determination of cloud-base properties. We then present an evaluation of its behaviour in the global model, with particular focus on the spatiotemporal distribution of clouds and precipitation.

## 2 Model description

### 2.1 The ECHAM–HAMMOZ composition–climate model

ECHAM6 (Roeckner et al., 2003; Stevens et al., 2013) is the sixth-generation climate model developed at the Max Planck Institute for Meteorology. It has a spectral dynamical core, solving prognostic equations for vorticity, divergence, surface pressure and temperature in spherical harmonics with a triangular truncation. A hybrid sigma/pressure vertical coordinate is used. Physical parameterisations are solved on a corresponding Gaussian grid. Tracer transport is semi-Lagrangian in grid-point

space (Lin and Rood, 1996).

HAM2 (Stier et al., 2005; Zhang et al., 2012) is a two-moment modal aerosol scheme based on the M7 framework (Vignati, 2004), representing five components (sulfate, sea salt, black carbon, particulate organic matter and mineral dust) in seven internally mixed log-normal modes (four soluble and three insoluble). ECHAM–HAMMOZ also includes the MOZ gas-phase chemistry model; however

this is not used in the present study.

In ECHAM–HAM, large-scale clouds follow the two-moment prognostic condensate scheme of Lohmann et al. (2007) with modifications by Lohmann and Hoose (2009). When running without HAM, ECHAM uses the Lohmann and Roeckner (1996) one-moment prognostic condensate scheme. In both cases cloud cover is diagnosed from relative humidity following Sundqvist et al.

(1989). Convection is parameterised by the bulk mass-flux scheme of Tiedtke (1989) with modifications by Nordeng (1994); we replace this with the Convective Cloud Field Model (described below) except in our control simulations.

The model version used here is ECHAM6.1–HAM2.2–MOZ0.9 (with and without the addition of CCFM) in its default ECHAM–HAM configuration at the commonly-used T63L31 resolution

($\sim 1.875°$ on 31 levels up to $10\,\mathrm{hPa}$ with a $2 \times 12$-minute leapfrog timestep), plus Abdul-Razzak and Ghan (2000) aerosol activation with an updraught velocity distribution for stratiform clouds derived from the boundary-layer turbulent kinetic energy (TKE) following West et al. (2014), and the model correspondingly retuned following the approaches outlined in Mauritsen et al. (2012). The simulations using ECHAM6.1 without HAM are at T63L47 (same tropospheric vertical resolution





but extended to $0.01\,\mathrm{hPa}$, with a $2 \times 10$-minute leapfrog timestep), as described in Stevens et al. (2013). The reason for this is that the supported resolutions for ECHAM and ECHAM–HAMMOZ differ, and using a supported choice for each ensures that both control simulations are comparable with those carried out elsewhere.

## 2.2 The Convective Cloud Field Model (CCFM)

CCFM is a spectral convective parameterisation representing the statistical effects of a heterogeneous ensemble of cumulus clouds based on Arakawa and Schubert (1974), extended with an explicit cloud model based on a one-dimensional steady-state entraining plume. These clouds interact with their grid-scale environment through entrainment and detrainment, and with one another via their effects on this common environment, as illustrated schematically in Figure 1. These interactions generate a

system of coupled linear first-order differential equations representing the competition for convective available potential energy (CAPE), which can be solved to determine the number of clouds of each type under the assumption of convective quasi-equilibrium.

An overview of CCFM is presented in the rest of this section; further details of the derivation and rationale can be found in Wagner and Graf (2010).

### 110 2.2.1 Entraining plume cloud model

Each cloud type which could exist in a particular grid cell is represented by a (vertical) one-dimensional Lagrangian entraining plume model. The cloud is assumed to be in a steady state on the scale of a host-model time step, and to have uniform properties over its horizontal cross-section. The cloud model is initiated at cloud base with a parcel of perturbed environmental air, which is diluted by

115 turbulent mixing entrainment through the lateral boundary of the cloud, and eventually detrained at cloud top.

The dynamical part of the model is formulated following Simpson and Wiggert (1969) and Kreitzberg and Perkey (1976), and solves the vertical momentum, thermodynamic and continuity equations to determine the evolution of vertical velocity $w$, temperature $T$ and cloud radius $r$ from cloud

base to cloud top (determined as the lowest level at which $w < w_\mathrm{min}$, set to $0.1\,\mathrm{m\,s^{-1}}$). The entrainment rate $\mu$ (with units of inverse length) is assumed to be inversely proportional to $r$:

$$\mu = \frac{C_\mu}{r};\tag{1}$$

the dimensionless constant of proportionality $C_\mu$ is set to 0.20 as in Wagner and Graf (2010).

This dynamical model is coupled to a microphysical parameterisation for the development of

125 liquid water, ice and precipitation, which is based on the one-moment bulk mixed-phase scheme used in ECHAM5 (Lohmann and Roeckner, 1996; Zhang et al., 2005).





### 2.2.2 Sub-cloud dry convection, triggering and activation

In Wagner and Graf (2010), cloud base was determined as the lifting condensation level (LCL) of a parcel lifted adiabatically from the lowest model level. The entraining plume was then initialised

at cloud base using environmental air with a fixed positive buoyancy perturbation. This approach is simple to implement, but has two main drawbacks: firstly, it does not consider the role of convective inhibition (CIN) whereby a thermodynamic inversion below the LCL prevents the development of convective clouds; secondly, it provides no information about cloud-base $w$ for calculating the activation of cloud condensation nuclei (CCN).

In the version used here, CCFM has been extended with a treatment of sub-cloud dry convection to address these points. This uses the same entraining plume model as described above, but with an unsaturated parcel of air from a configurable level near the surface (again with a fixed positive buoyancy perturbation: $w = 1\,\mathrm{m\,s^{-1}}, T = T_{\mathrm{LS}} + 2.8\,\mathrm{K}, q = q_{\mathrm{LS}} + 1 \times 10^{-4}\,\mathrm{kg\,kg^{-1}}$). If the plume reaches a level at which condensation occurs, this is determined to be the cloud base. If $w$ drops

below $w_{\mathrm{min}}$ before this happens, no cloud is formed.

The exact magnitudes of these perturbations are poorly constrained, and it is anticipated that a future physically-based approach will take account of orographic variability, surface type and boundary-layer structure. In the present scheme, however, the $T$ perturbation has the dominant effect, and this is tuned to ensure that the model remains close to radiative balance without re-tuning

other components of the model compared to the simulations with Tiedtke–Nordeng.

The sub-cloud model is run for a range of $n_{\mathrm{sub}}$ (set to 20) initial parcel radii, linearly spaced from 200 m up to the diagnosed depth of the planetary boundary layer ($z_{\mathrm{PBL}}$). Cloud base is determined by the first (i.e. smallest) of these to produce a cloud. If none of these parcels is able to produce a cloud, due to strong CIN, no convection is simulated for this grid column.

The potential cloud types for which the actual cloud model is run are defined by linearly spacing $n_{\mathrm{cld}}$ (set to 10) cloud-base radii from $r_1$ to $\max(r_{\mathrm{max}}, z_{\mathrm{PBL}})$ where $r_1$ is the cloud-base radius of the first sub-cloud parcel to condense, and $r_{\mathrm{max}}$ that of the largest cloud produced, at the cloud base level. The initial parcel properties ($w, T, q$) for each cloud type are determined by linearly interpolating in $r$ from the cloud-base properties of the sub-cloud parcels. The cloud-base $w$ determined in this way

is then used to drive the Abdul-Razzak and Ghan (2000) activation scheme to determine the cloud droplet number concentration (CDNC) based on aerosol entrained from the cloud base level. Each cloud type has its own vertical velocity and CDNC, which will have an impact on the microphysics and hence (along with the differing entrainment rates) on the development of the cloud and its effect on the resolved scale via heating, drying, precipitation and detrainment.



### 2.2.3 Determining the cloud spectrum: Interactions between clouds and their environment

Convective clouds in CCFM interact with their environment via environmental controls on the buoyancy of the rising parcel, entrainment of environmental air (with its heat, moisture and aerosol content) into the convective plumes through mixing at the cloud edge, and detrainment of the air in the convective plume into the environment at cloud top. There is also a small downward motion, or compensating subsidence, in the portion of each grid box not covered by convective plumes, such that the parameterisation is locally mass conserving.

Through these effects, the environment controls the profile of each convective plume, but the plumes in turn modify their environment in particular through changes in temperature and humidity during detrainment which alter the thermodynamic profile of the column. This can be expressed in terms of the cloud work function (CWF) introduced by Arakawa and Schubert (1974), defined as

$$\underbrace{A\left(T_{v,i}, T_{v,env}\right)}_{= A_i} = \frac{1}{w_{b,i} r_{b,i}^2 \rho_{b,i}} \int_{z_{base,i}}^{z_{top,i}} \frac{T_{v,i} - T_{v,env}}{T_v^{env}} g w_i r_i^2 \rho_i \, \mathrm{d}z, \tag{2}$$

where $w_{b,i}$, $r_{b,i}$ and $\rho_{b,i}$ are the vertical velocity, radius and density at the base of cloud type $i$ (as obtained from the sub-cloud model), and $T_{v,i}$ and $T_{v,env}$ are the virtual temperature in the cloud model and grid-box environment respectively.

Under assumptions of convective quasi-equilibrium as discussed in Wagner and Graf (2010), where more detail of the derivation may be found, the number of clouds of each type evolves following:

$$\begin{aligned}
\frac{\mathrm{d}n_i}{\mathrm{d}t} &= \frac{n_i}{A_i} \frac{\mathrm{d}A_i}{\mathrm{d}t} \\
&= \frac{n_i}{A_i} \left[ \underbrace{\left(\frac{\mathrm{d}A_i}{\mathrm{d}t}\right)_{ls}}_{= F_i} + \sum_{j=1}^{n_{cld}} \underbrace{\left(\frac{\mathrm{d}A_i}{\mathrm{d}t}\right)_j}_{= n_j k_{ij}} \right],
\end{aligned} \tag{3}$$

where $n_i$ is the number of clouds of type $i$ per unit horizontal area.

The terms on the right represent the production of CAPE by the large-scale environment and the suppression of clouds of type $i$ by those of type $j$ respectively. The "kernel" $k_{ij}$ represents the effect of a single cloud of type $j$ per unit area on those of type $i$ in the same GCM column.

These interactions give rise to a Lotka–Volterra system of coupled first-order differential equations for the evolution of the number of clouds of each type based on their competition for CAPE:

$$\frac{\mathrm{d}n_i}{\mathrm{d}t} = f_i n_i \left( 1 - \sum_{j=1}^{n} a_{ij} n_j \right), \tag{4}$$

where the coefficients $f_i = F_i / A_i$ and $a_{ij} = -k_{ij} / F_i$. When integrated forward to equilibrium, determining the number of clouds of each type present, this equation forms the closure for CCFM. This requires knowledge of the forcing and interaction coefficients, which are determined by making use



of the model's operator splitting to separately calculate the change in the CWF due to large-scale processes, and due to a single cloud of each type in isolation. In the notation of (2),

$$F_i = \frac{A\left(T_{v,i}, T_{v,\mathrm{env+ls}}\right) - A\left(T_{v,i}, T_{v,\mathrm{env}}\right)}{\Delta t} \tag{5}$$

$$k_{ij} = \frac{A\left(T_{v,i}, T_{v,\mathrm{env+}j}\right) - A\left(T_{v,i}, T_{v,\mathrm{env}}\right)}{\Delta t} \tag{6}$$

where $T_{v,\mathrm{env}}$ refers to the virtual temperature of the environment at the start of the timestep, $T_{v,\mathrm{env+ls}}$

that when updated due to the large-scale processes only, $T_{v,\mathrm{env+}j}$ its value when updated due to a single cumulus cloud of type $j$, and $\Delta t$ is the GCM timestep.

The Lotka–Volterra equations (4) are integrated using an explicit fourth-order Runge–Kutta method with an adaptive step size, until the $n_i$ converge or a limit of 1000 s or 1000 steps is reached (which happens only rarely, in particularly stiff cases, and does not appear to have a significant impact on

the overall results).

The modification of the large-scale environment by convective heating/cooling and drying/moistening due to clouds of each type is calculated following Tiedtke (1989) (extended to include ice-phase transitions):

$$\left(\frac{\partial \bar{s}}{\partial t}\right)_{\mathrm{cu}} = L_v\left(C - E\right) + L_f\left(F - M\right) - \frac{1}{\rho}\frac{\partial}{\partial z}\left(\overline{\rho w' s'}\right) \tag{7}$$

$$\left(\frac{\partial \bar{q}_v}{\partial t}\right)_{\mathrm{cu}} = \left(C - E\right) - \frac{1}{\rho}\frac{\partial}{\partial z}\left(\overline{\rho q_v' s'}\right), \tag{8}$$

where $s$ is the dry static energy, $L_v$ and $L_f$ are the latent heat of vaporisation and fusion, $q_v$ is the water vapour mixing ratio, $(C - E)$ is the net condensation rate and $(F - M)$ the net freezing rate (vapour–ice transitions are included in both, as though via the liquid phase). Overbars $(\bar{\cdot})$ denote grid-scale horizontal means, while primes $(\cdot')$ denote local deviations due to the convective clouds

parameterised by CCFM.

Expanding the latent-heating and sub-grid transport terms on the right-hand side of (7) and (8) in terms of the convective mass flux, and changing to pressure coordinates assuming hydrostatic balance, leads to

$$\left(\frac{\partial \bar{s}}{\partial t}\right)_{\mathrm{cu}} = g\frac{\partial}{\partial p}\sum_{j=1}^{n_{\mathrm{cld}}} M_j\Big[s_j - \bar{s} - L_v\left(q_{l,j} + q_{r,j}\right) \tag{9}$$

$$- L_f\left(q_{i,l,j} + q_{s,l,j}\right) - L_s q_{i,v,j}\Big]$$

$$\left(\frac{\partial \bar{q}_v}{\partial t}\right)_{\mathrm{cu}} = g\frac{\partial}{\partial p}\sum_{j=1}^{n_{\mathrm{cld}}} M_j\Big[q_{v,j} - \bar{q}_v + q_{l,j} + q_{r,j} + q_{i,j} + q_{s,j}\Big]. \tag{10}$$

The effect on any other physical quantity $\phi$, e.g. tracers or momentum, is similarly given by

$$\left(\frac{\partial \bar{\phi}}{\partial t}\right)_{\mathrm{cu}} = g\frac{\partial}{\partial p}\sum_{j=1}^{n_{\mathrm{cld}}} M_j\Big[\phi_j - \bar{\phi} + S_{\phi,j}\Big]. \tag{11}$$

where $S_{\phi,j}$ represents the net source of $\phi$ within a cloud of type $j$.



Finally, the precipitation rate is calculated as the vertically-integrated rate of rain and snow production within each cloud; the cloud-top detrainment rate of water vapour, liquid water, ice and other tracers is simply the updraught flux of that quantity at cloud top.

## 3 Method

In order to evaluate the performance of CCFM in the global model, we have conducted several one-year (plus 3 months' spin-up) free-running simulations using ECHAM–HAM with CCFM in different configurations, as well as a corresponding reference simulation using the standard Tiedtke–Nordeng scheme. These configurations are listed in Table 1, and vary in the vertical level at which the sub-cloud dry convection model is initiated, a parameter to which the triggering of convection turns out to be quite sensitive. These vary from L$-$0 (lowest model level, $\sim 30\,\mathrm{m}$ above the surface) to L$-$3 (three levels higher, $\sim 600\,\mathrm{m}$ above the surface).

For the best-performing configuration (L$-$2) we have conducted a 30-year AMIP-type simulation, along with an equivalent simulation using Tiedtke–Nordeng. These AMIP-type simulations have been conducted both with ECHAM–HAM and standard ECHAM (i.e. without HAM) to allow comparison across configurations of the host model. For the ECHAM–HAM simulations, aerosol and precursor emissions for the present day (i.e. year 2000) are used as per the AeroCom Phase II/ACCMIP recommendations (http://aerocom.met.no/emissions.html). For simulations without HAM, the MAC aerosol climatology (Kinne et al., 2013) is used.

We analyse these in terms of the annual mean geographical distribution of column properties (liquid and ice water paths, cloud cover and surface precipitation) and the meridional–vertical distribution of zonal-mean local properties (liquid and ice water contents and cloud fraction). We also look at the annual mean top-of-atmosphere (TOA) cloud radiative effect (CRE) and net radiative flux.

Surface precipitation is evaluated against a monthly climatology from the Global Precipitation Climatology Project (GPCP; Adler et al., 2003; Huffman et al., 2009). Cloud cover is evaluated against a monthly climatology derived from the CALIPSO–GOCCP (Chepfer et al., 2010) data set, using the CFMIP Observational Simulator Package (COSP; Bodas-Salcedo et al., 2011). (This is the grid-scale cloud cover diagnosed based on the total relative humidity including any contribution from moisture detrained from the convective parameterisation; the explicit area coverage of the actual convective updraughts represented by CCFM is negligible in comparison.) CRE and radiative flux are evaluated against the CERES–EBAF (Loeb et al., 2009) data set. These evaluations are carried out both visually via annual-mean difference plots, and statistically via Taylor (2001) diagrams.

The seasonal and diurnal cycles of precipitation are also studied in three specific regions of convective activity: the Amazon (45–65°W, 15°S–5°N), the Congo (10–30°E, 11°S–7°N) and Indonesia (105–125°E, 10°S–10°N). These are evaluated against the Tropical Rainfall Measuring Mission



(TRMM) 3B42 merged precipitation data set (TRMM, 2011) over ten years of overlap with the
AMIP simulations (1999–2008).

## 4 Results and discussion

### 4.1 Hydrological fields

Figure 2 shows the annual mean column-integrated liquid and ice water paths, (2D) cloud cover and
surface precipitation from ECHAM(–HAM) using both CCFM (L−2 configuration) and Tiedtke–
Nordeng convection. The geographical patterns are broadly similar, although there is generally less
liquid and ice when CCFM is used, both in the tropics and the mid-latitudes.

Figure 3 shows the annual and zonal mean meridional–vertical profiles of liquid and ice content
and (3D) cloud fraction from these two simulations. The generally lower liquid and ice contents
using CCFM are again apparent, with the strongest difference being in the tropical lower troposphere,
where there is very little liquid water when using CCFM. This may be related to the use of cloud-
edge mixing detrainment from deep convection in the bulk mass-flux formulation, allowing liquid
water to detrain out of the lower part of such clouds, while CCFM detrains only at the explicit top of
each cloud type.

It is important to note, however, that the differences in these fields from the choice of convection
scheme are not as great as those between ECHAM and ECHAM–HAM, although the spatial signa-
tures are different. ECHAM–HAM generally has more liquid and less ice than standard ECHAM,
especially in the mid-latitudes; this is most likely due to their different large-scale cloud schemes as
well as different tuning choices.

### 4.2 Evaluation against observations

#### 4.2.1 Precipitation and cloud vs. GPCP and CALIPSO

In order to evaluate the impact of CCFM on precipitation and cloudiness in the model, Figure 4
shows the difference between the annual mean surface precipitation and (COSP-simulated CALIPSO-
like) cloud cover from ECHAM–HAM with Tiedtke–Nordeng and CCFM, and GPCP and CALIPSO–GOCCP
climatologies respectively. The precipitation differences show very similar patterns with both con-
vection schemes, suggesting that these may be constrained by larger-scale processes within the
model or underlying assumptions common to both schemes. In the case of cloud cover, however,
the patterns are different: CCFM shows a cloudy bias over the western side of the ocean basins,
while Tiedtke–Nordeng shows a clear bias over the eastern side. Although the geographical pat-
terns of bias are different, neither is obviously better. The corresponding results for the ECHAM
simulations without HAM are qualitatively similar (not shown).





The cloud cover is quite sensitive to the model level at which the sub-cloud dry convection is initiated. Choosing two levels above the lowest ($\sim 350\,\mathrm{m}$, L−2 configuration) produces the smallest overall bias, and this is our "standard" configuration used elsewhere in this paper. The difference between simulated cloud cover using different initiation levels and CALIPSO–GOCCP is shown in

Figure 5. Choosing a lower level (L−1 or L−0) produces too little cloud, particularly in regions of marine stratocumulus, perhaps due to suppression by low-level inversions. Choosing a higher level (L−3) produces too much cloud, similar to what happens when our new sub-cloud model is not used (not shown). Increasing/decreasing the temperature perturbation has a similar (but lesser) effect to raising/lowering the initiation level (see Figure 6). Choosing $2.8\,\mathrm{K}$ minimises the cloud cover bias in

the L−2 configuration and keeps the model close to radiative balance, as mentioned in Section 2.2.2. The spatial distribution of precipitation, however, is relatively robust against changes to the initiation level and temperature perturbation.

The comparison between the various model configurations and observations is illustrated statistically in Figure 7 with Taylor (2001) diagrams of the monthly climatology of per-grid-point precipi-

300 tation, 2D cloud cover and 3D cloud fraction. In ECHAM–HAM, CCFM improves the precipitation distribution compared to Tiedtke–Nordeng, in terms of both variability and root-mean-square error (RMSE), and slightly in terms of correlation, at the expense of a slightly increased bias. The improved distribution is almost as good as that from ECHAM without HAM (which has been more extensively tuned, and where there is little difference based on the convection scheme). It is possible

that, with suitable tuning, ECHAM–HAM with Tiedtke–Nordeng would perform as well – though this might come at the cost of introducing the less realistic diurnal cycle seen in standard ECHAM (see Section 4.2.3).

For 2D cloud cover, the correlation does worsen when CCFM is used in its L−2 configuration, although the bias and variability are improved. A strong sensitivity to initiation level (and to a lesser

extent the magnitude of the temperature perturbation) is apparent, however, with L−0, L−1 and L−3 all exhibiting lower correlations and large biases (see Figure S1 in the supplement) matching the effects visible in Figure 5. For 3D cloud fraction, the difference between ECHAM with and without HAM is larger than that due to the choice of convection scheme: ECHAM–HAM shows poorer correlation while standard ECHAM has greater bias and excess variability. The smaller additional signal

from the convection scheme is similar to that for the 2D cloud cover. It is probably not the HAM aerosol scheme itself that makes the difference, but rather the switch from one-moment Lohmann and Roeckner (1996) to two-moment Lohmann et al. (2007) microphysics and associated re-tuning of the model.

### 4.2.2 Radiative effects vs. CERES

The annual mean net downward radiative flux at the top of the atmosphere (TOA) and cloud radiative effect (CRE) simulated in ECHAM–HAM using CCFM (L−2 configuration) and Tiedtke–Nordeng





convection are compared with a CERES–EBAF climatology in Figure 8. The split between short-wave and long-wave effects can be found in the supplement as Figure S2. The main change between Tiedtke–Nordeng and CCFM appears to be the shift from a dipole in the tropics (with negative bias in the northern tropics and a positive bias in the south) to a negative tropical bias balanced in the mid-latitudes. However, the difference between the convective parameterisations appears no greater than that between ECHAM–HAM and ECHAM (not shown). The corresponding Taylor (2001) diagrams in Figure 9 confirm that the L−2 configuration is close to Tiedtke–Nordeng in both ECHAM and ECHAM–HAM overall, although the SW and LW CRE are overly strong but mostly cancel. These are aspects that are very sensitive to tuning of the large-scale cloud scheme, and it is likely that re-tuning without the constraint that both Tiedtke–Nordeng and CCFM should be in balance with the same parameter values would yield an improvement here. The other CCFM configurations perform significantly worse (see Figure S3 in the supplement), particularly in terms of bias (because they are out of radiative balance) and excess variability in either then long-wave or short-wave CRE.

### 4.2.3 Seasonal and diurnal cycles vs. TRMM

To assess the seasonal cycle of convective activity, the top row of Figure 10 shows the monthly mean fraction of total annual surface precipitation from the ECHAM–HAM AMIP simulations in the Amazon, Congo and Indonesia regions against that from the TRMM 3B42 merged precipitation data set, over a ten-year overlap period (1999–2008). In the Amazon and Congo regions, both Tiedtke–Nordeng and CCFM (L−2) capture the seasonal cycle reasonably well. The seasonal cycles from the alternative CCFM configurations differ by less than the interannual variability in L−2, so no clear distinction can be inferred from their seasonal cycles. In Indonesia, however, Tiedtke–Nordeng appears to capture the seasonal cycle better, and the alternative CCFM configurations differ much more markedly. In ECHAM without HAM, however, neither scheme captures the seasonal cycle in Indonesia well (Figure 11), suggesting that this region is highly sensitive to the tuning of both convective parameterisations.

The diurnal cycles vary considerably from month to month; those for March and August are shown in the lower part of Figure 10 as a representative selection and the full set is included in the supplement. The cycles are normalised to show the fraction of mean daily precipitation at each (local) time of day. Neither scheme reliably captures both the magnitude and timing of the diurnal cycle well, which is a persistent problem in convective parameterisation in low-resolution climate models; however in general CCFM appears to do so as well as or better than Tiedtke–Nordeng, especially in terms of timing. The interannual variability is quite consistent between both models and observations. The differences between CCFM configurations become more significant, suggesting that the treatment of convective initiation is likely to be a key process for further improvement in the diurnal cycle.





Figure 11 shows the equivalent for ECHAM running without HAM. In this case, CCFM behaves similarly to in ECHAM–HAM, while Tiedtke–Nordeng has an overly strong diurnal cycle in both the Amazon and Congo regions, which also peaks too early in the day. As noted above, Tiedtke–Nordeng also fails to capture the seasonal cycle in Indonesia in this configuration. This strong difference in the behaviour of the Tiedtke–Nordeng scheme between ECHAM and ECHAM–HAM may be related to their use of quite different values of its parameters for climatological tuning, resulting in different physical behaviour on shorter timescales.

### 4.2.4 Updraught velocity, area and cloud-top pressure distributions

One of the unique features of CCFM is its ability to determine the distribution of cloud sizes and updraught velocities in a given grid-scale environment, making it suitable for the study of both convective-cloud microphysics and aerosol effects and cloud-field morphology. Figure 12a shows the annual and global joint distribution of cloud-base radius and updraught velocity from the simulation using CCFM (L−2 configuration). There is a tendency for broader-based clouds to have stronger updraughts, but a large and bimodal variability in the simulated velocity at any given radius, which we would expect to translate into significant variability in the activation of aerosol into cloud droplets.

We can also obtain the joint distribution of the maximum radius reached by the updraught in each column, and the pressure at its cloud top, shown in Figure 12b. Again, there is some correlation with broader clouds tending to be deeper, but significant variability around this, opening the way to investigate the impact of aerosol or other climate forcings on cloud field morphology.

There is potential for evaluating these distributions against both convection-resolving simulations and observations in future studies, although the sources of suitable data are still quite limited and there are many challenges to overcome in conducting a like-for-like comparison of convective cells between such different representations.

A promising approach here is to evaluate single-column model simulations against ground-based radar observations. An upcoming study will compare CCFM vertical velocity and mass-flux profiles with radar retrievals at Darwin, Australia (Collis et al., 2013; Kumar et al., 2015). Convective vertical velocities are essential for convective microphysics and aerosol-convection interaction and hence, as highlighted by Donner et al. (2016), their accurate representation may be important for climate sensitivity and future climate projections.

### 5 Conclusions

We have introduced the Convective Cloud Field Model (CCFM) as a component of the ECHAM6–HAM2 global model. Unlike the usual bulk mass-flux parameterisation (Tiedtke–Nordeng), this is able to dynamically represent a heterogeneous ensemble of convective clouds within the GCM grid column,



allowing a representation of cloud-field morphology with a diversity of both cloud-scale properties and microphysical processes within the ensemble. These capabilities make the model particularly well suited to capturing the interactions between aerosol and convection at the global scale, filling a gap between high-resolution models where convection is explicit rather than parameterised (but

which cover limited domains), and typical global models whose parameterisations cannot capture the sub-grid-scale processes on which such interactions depend.

We have evaluated the performance of CCFM against remote-sensing observations of both cloud and precipitation at the global scale, and also seasonal and diurnal cycles at the regional scale. With suitably-chosen parameters, CCFM gives an improved spatiotemporal distribution of precipitation

in ECHAM–HAM compared to Tiedtke–Nordeng, including improved timing of the diurnal cycle, and performs almost as well in terms of cloud fraction and radiative effects even without re-tuning of other components of the model. This is in keeping with the results seen by Wagner and Graf (2010) in single-column model studies with an earlier version of the model.

Both cloud fraction and the diurnal cycle of precipitation are sensitive to the way convective

triggering is handled by the sub-cloud dry convection. An improved physical basis for the choice of initiating perturbations might lead to a better representation of the diurnal cycle, and reduce the need for tuning based on cloud fraction.

Given that its representation of cloud and precipitation fields is at least as good as the standard scheme, but provides the cloud-base vertical velocity required to diagnose aerosol activation, and

410 the area coverage required to represent cover/lifetime effects, we conclude that CCFM is ready to be used to investigate many of the aerosol indirect effects on convective cloud fields. Further development of the microphysics to use a multi-moment mixed-phase scheme will allow this to be extended to cover additional proposed effects related to the ice particle size distribution.

*Acknowledgements.* This research is funded from the European Research Council under the European Union's

Seventh Framework Programme (FP7/2007–2013) / ERC grant agreement no. FP7–280025 (ACCLAIM). The research leading to these results has received funding from the European Union's Seventh Framework Programme (FP7/2007-2013) project BACCHUS under grant agreement no. 603445. The ECHAM–HAMMOZ model is developed by a consortium composed of ETH Zürich, Max Planck Intitut für Meteorologie, Forschungzentrum Jülich, the University of Oxford and the Finnish Meteorological Institute, and managed by the Center for

Climate Systems Modeling (C2SM) at ETH Zürich. This work used the ARCHER UK National Supercomputing Service (http://www.archer.ac.uk/).



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





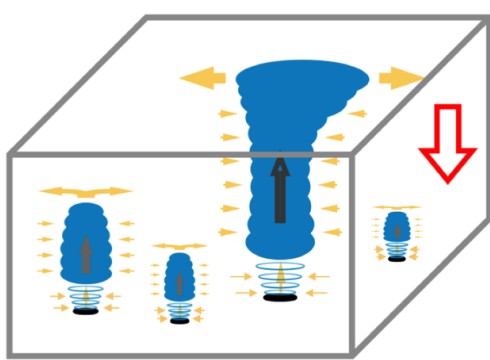

**Figure 1.** Illustration of the heterogeneous convective clouds represented by CCFM in a GCM grid box, including the newly-included sub-cloud dry convection.



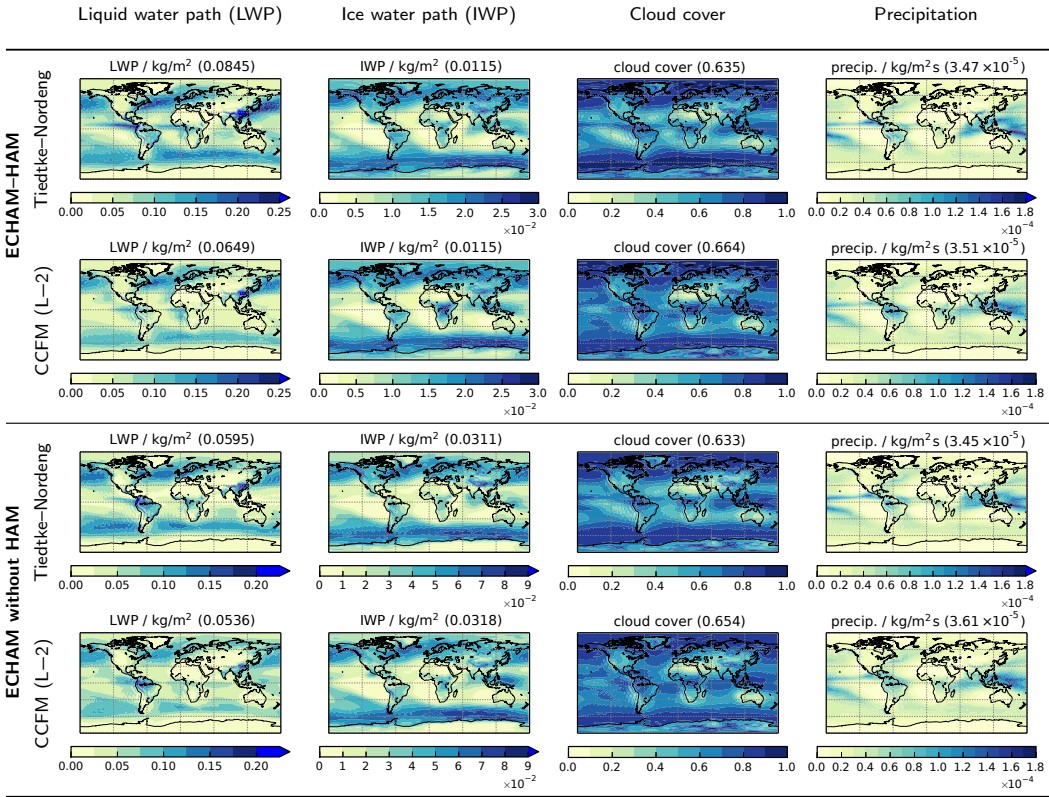

**Figure 2.** Annual mean (from left to right) liquid water path (LWP), ice water path (IWP), cloud cover and surface precipitation from 30-year AMIP-type simulations using ECHAM(–HAM) with Tiedtke–Nordeng and CCFM (L–2) convection. Note that the LWP and IWP scales are different for ECHAM and ECHAM–HAM due to their quite different magnitudes. (The numbers in parentheses show the annual global mean of each quantity.)




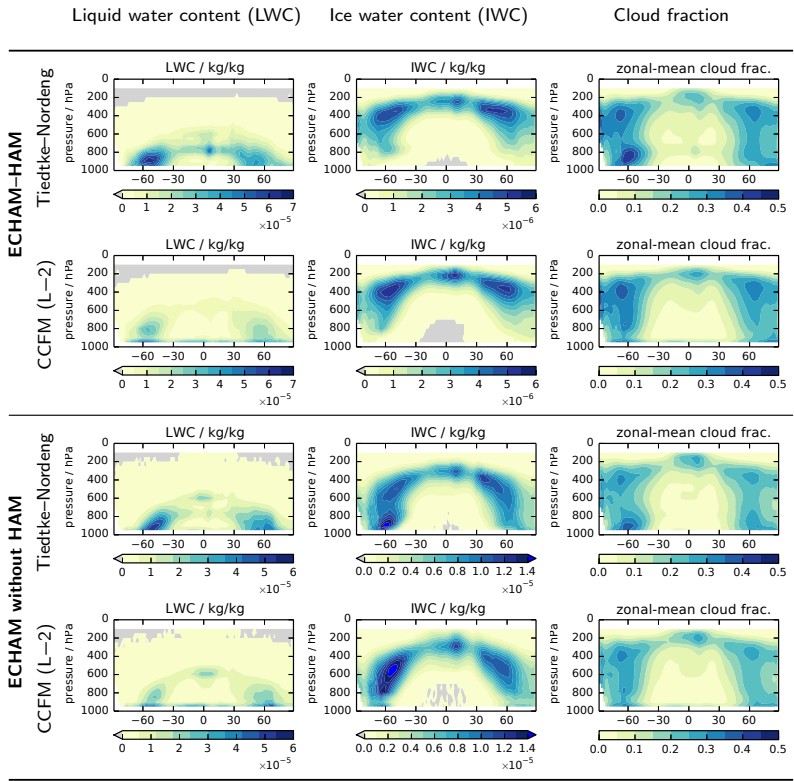

**Figure 3.** Annual and zonal mean (from left to right) liquid water content (LWC), ice water content (IWC) and cloud fraction from 30-year AMIP-type simulations using ECHAM(–HAM) with Tiedtke–Nordeng and CCFM (L−2) convection. Note that the LWC and IWC scales are different for ECHAM and ECHAM–HAM due to their quite different magnitudes.





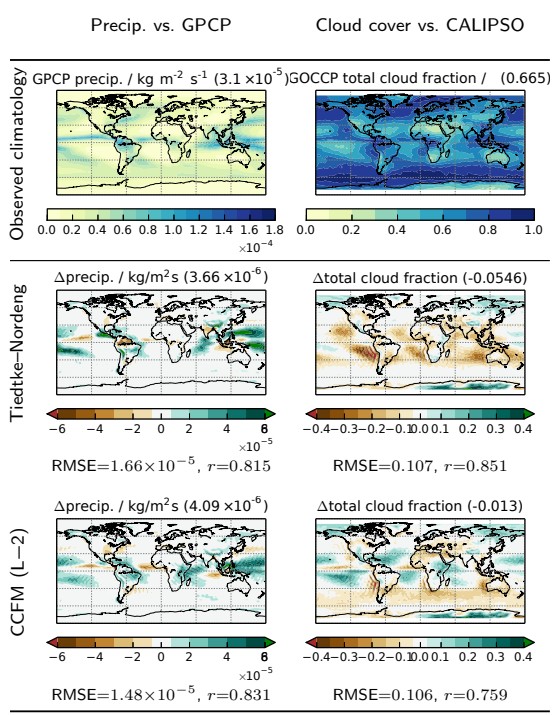

**Figure 4.** Difference in annual mean precipitation (left) and COSP-simulated cloud fraction (right) between 30-year AMIP-type simulations using ECHAM–HAM with Tiedtke–Nordeng and CCFM (L−2) convection, and the Global Precipitation Climatology Project (GPCP) and CALIPSO–GOCCP respectively.





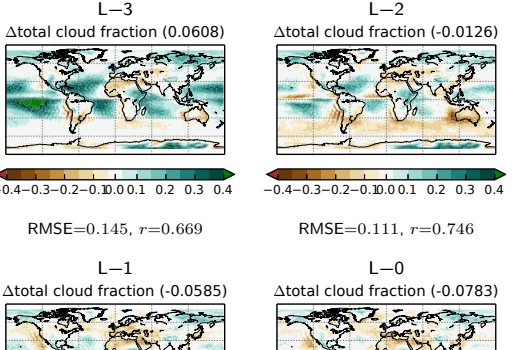

**Figure 5.** Difference in annual mean COSP-simulated cloud fraction between one-year simulations using ECHAM–HAM with CCFM in each configuration and CALIPSO–GOCCP.



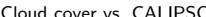

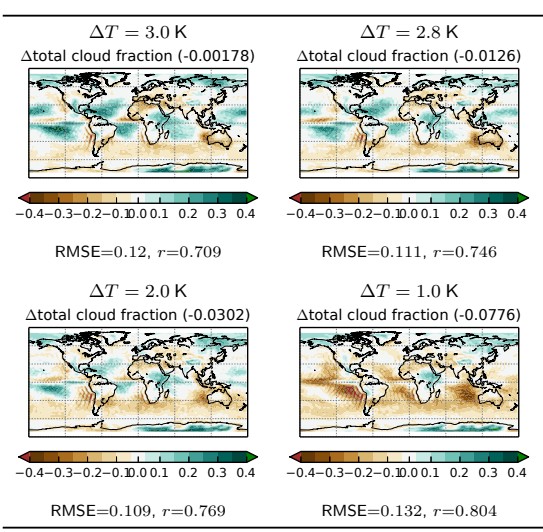

**Figure 6.** Difference in annual mean COSP-simulated cloud fraction between one-year simulations using ECHAM–HAM with CCFM (L−2) and CALIPSO–GOCCP, as a function of the temperature perturbation used to initiate the sub-cloud model.





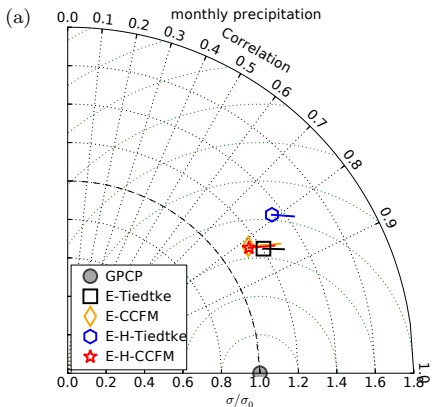

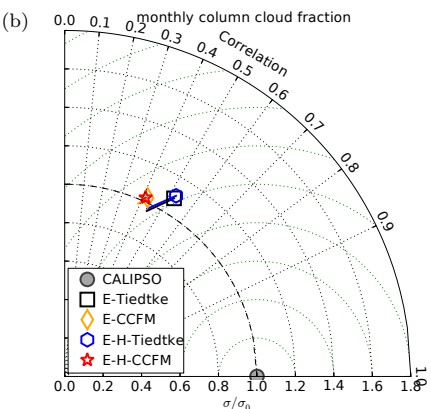

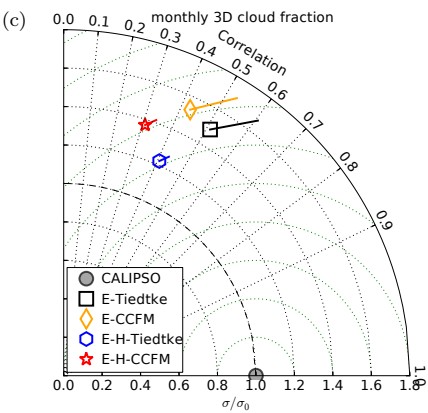

**Figure 7.** Taylor diagrams comparing (a) monthly mean precipitation, (b) COSP-simulated column cloud fraction and (c) COSP-simulated 3D cloud fraction (bottom) between 30-year AMIP-type simulations using ECHAM(–HAM) with Tiedtke–Nordeng and CCFM (L−2) convection, and the Global Precipitation Climatology Project (GPCP) and CALIPSO–GOCCP respectively. The line segments extending from each point indicate the normalised mean bias, as suggested in Taylor (2001).





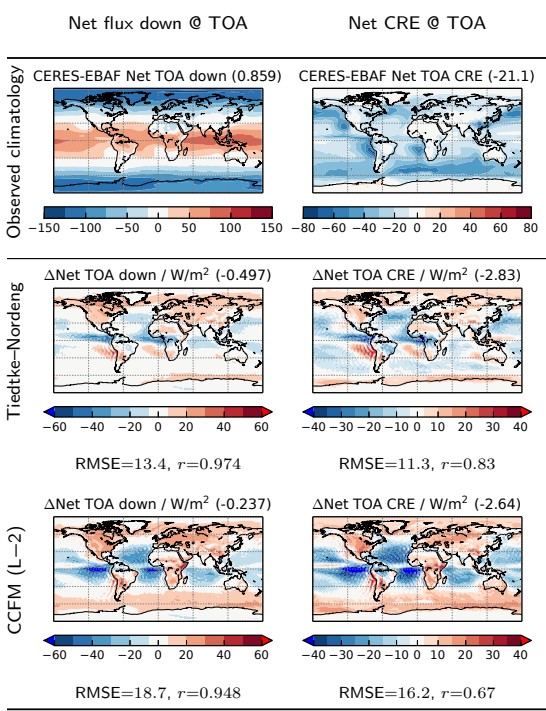

**Figure 8.** Difference in net downward radiative flux (left) and cloud radiative effect (right) at the top of the atmosphere between 30-year AMIP-type simulations using ECHAM–HAM with Tiedtke–Nordeng and CCFM (L−2) convection, and a CERES–EBAF climatology.





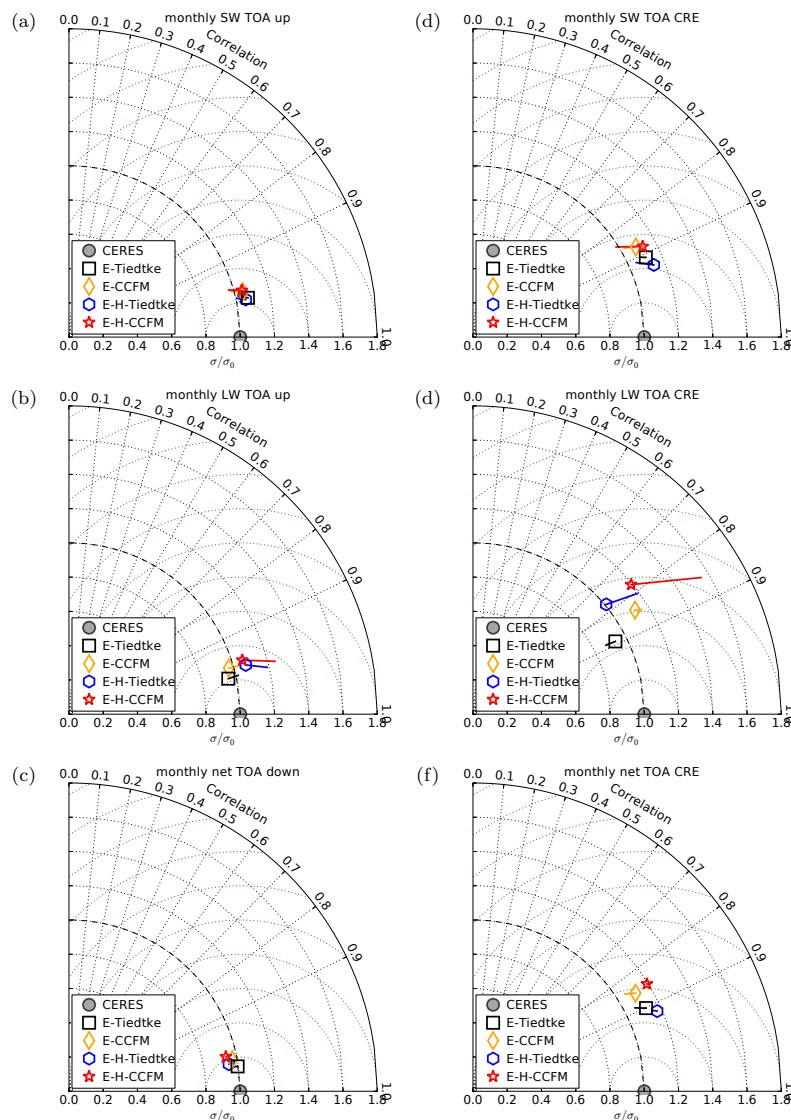

**Figure 9.** Taylor diagrams comparing monthly mean short-wave (a), long-wave (b) and net (c) radiative fluxes (left), and corresponding cloud radiative effects (d–f, right) at the top of the atmosphere between 30-year AMIP-type simulations using ECHAM(–HAM) with Tiedtke–Nordeng and CCFM (L−2) convection, and a CERES–EBAF climatology. The line segments extending from each point indicate the mean bias, as suggested in Taylor (2001).



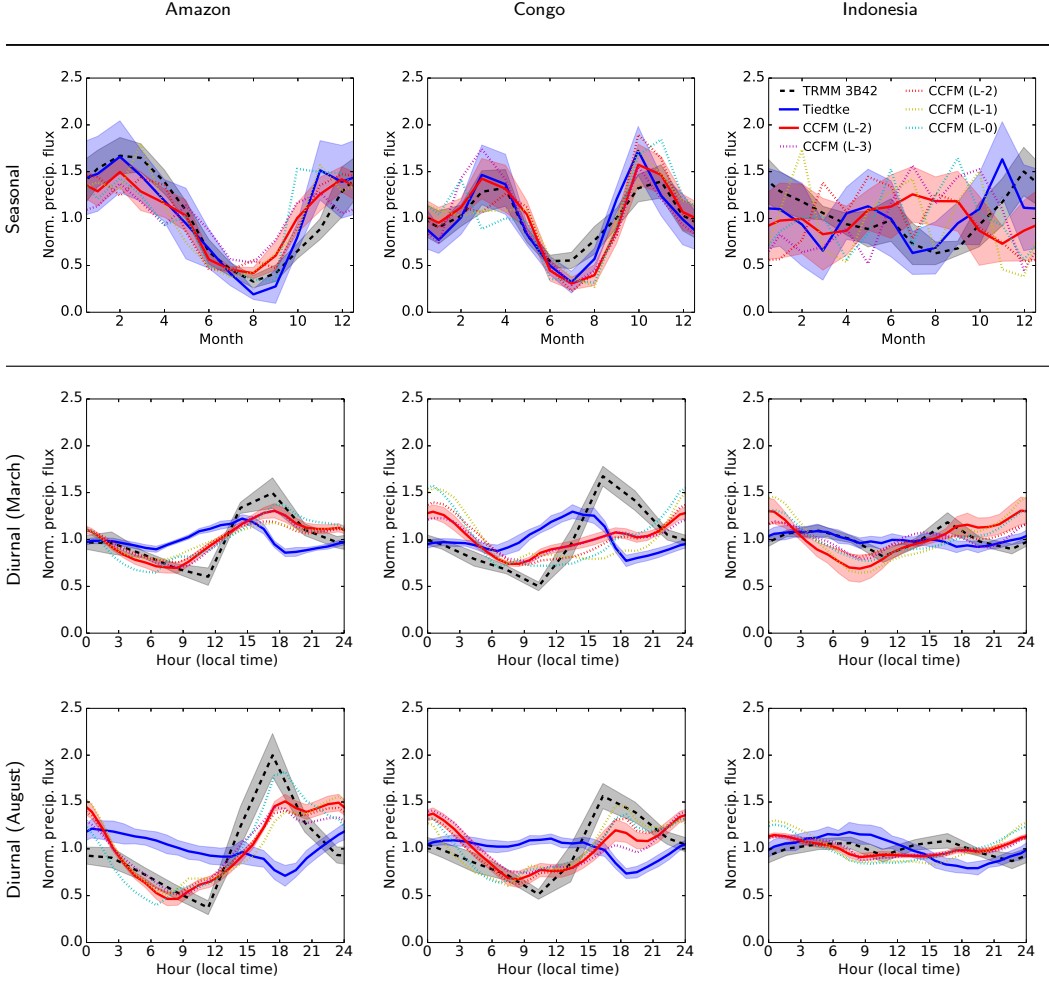

**Figure 10.** Normalised seasonal (top) and diurnal (below) cycles of precipitation in the Amazon (left), Congo (centre) and Indonesia (right) regions from a ten-year overlap between the TRMM 3B42 product and AMIP-type simulations using ECHAM–HAM with Tiedtke–Nordeng and CCFM (L−2) convection. The shaded regions indicate the interannual standard deviation of each data set. The dotted lines show the cycles from one-year simulations using alternative CCFM configurations. The diurnal cycles are in the local time of each region, and are shown for March and August; the full set of months is included in the supplement as Figures S4–6.



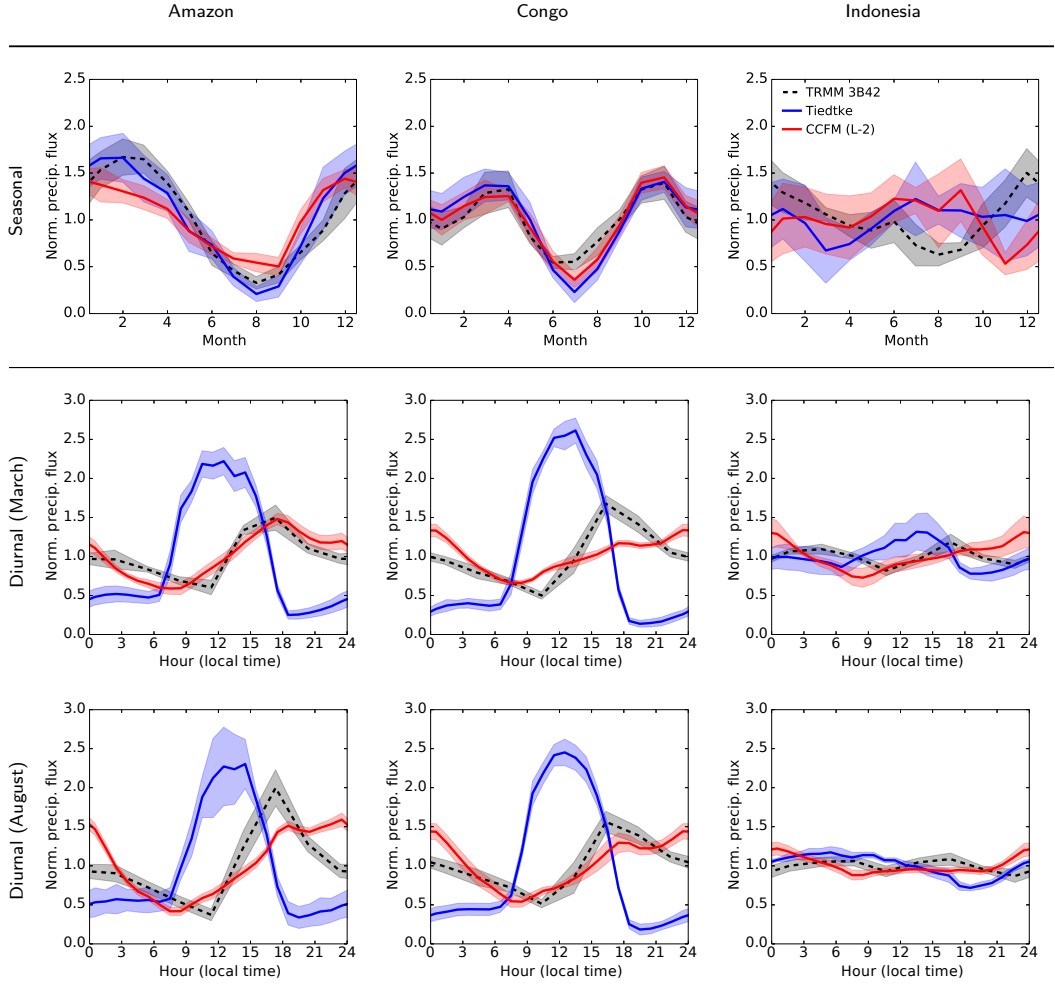

**Figure 11.** Normalised seasonal (top) and diurnal (below) cycles of precipitation in the Amazon (left), Congo (centre) and Indonesia (right) regions from a ten-year overlap between the TRMM 3B42 product and AMIP-type simulations using ECHAM (without HAM) with Tiedtke–Nordeng and CCFM (L−2) convection. The shaded regions indicate the interannual standard deviation of each data set. The dotted lines show the cycles from one-year simulations using alternative CCFM configurations. The diurnal cycles are in the local time of each region, and are shown for March and August; the full set of months is included in the supplement as Figures S4–6.





(a)

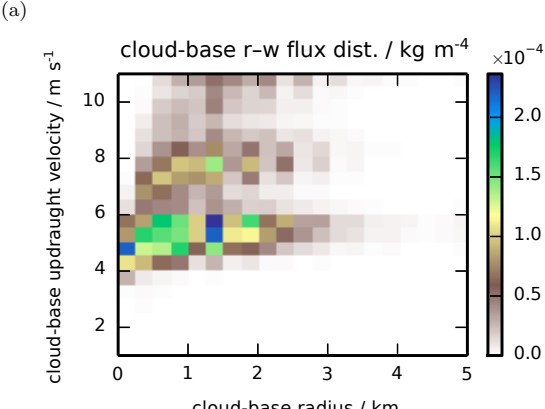

(b)

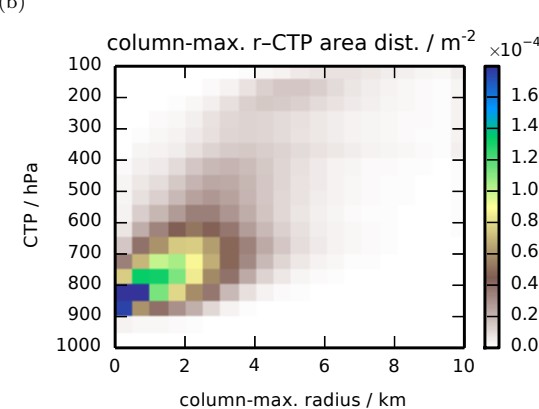

**Figure 12.** Joint distributions of (a) cloud-base radius and updraught velocity, and (b) column-maximum updraught radius and cloud-top pressure from a 30-year AMIP-type simulations using ECHAM–HAM with CCFM (L–2).



**Table 1.** ECHAM–HAM configurations

| Label | Convection scheme |
|---|---|
| **Tiedtke** | Standard Tiedtke–Nordeng scheme |
| **CCFM (L−3)** | CCFM, initiated 3 levels above lowest ($\sim 600\,\mathrm{m}$) |
| **CCFM (L−2)** | CCFM, initiated 2 levels above lowest ($\sim 350\,\mathrm{m}$) |
| **CCFM (L−1)** | CCFM, initiated 1 level above lowest ($\sim 150\,\mathrm{m}$) |
| **CCFM (L−0)** | CCFM, initiated at lowest level ($\sim 30\,\mathrm{m}$) |