# Peer review of "Dynamic sub-grid heterogeneity of convective cloud in a global model: Description and Evaluation of the Convective Cloud Field Model (CCFM) in ECHAM6–HAM2"

_Atmospheric Chemistry and Physics, 2016_

## Referee Comment (RC1) · Anonymous Referee #2 · 1 Sep 2016

**Referee comment on "Dynamic sub-grid heterogeneity of convective cloud in a global model: Description and evaluation of the Convective Cloud Field Model (CCFM) in ECHAM6-HAM2" by Kipling, Stier, Labbouz and Wagner**

**General comments**

The authors of this manuscript expand and test the cumulus convection scheme that is based on the multi-plume scheme from Arakawa and Schubert (1974) as implemented by Wagner and Graf (2010, CCFM). In particular a sub-cloud parcel description is included as well as cloud droplet number concentrations for each updraft. Parameter sensitivity experiments are documented and the simulations are compared to observations. The diurnal cycle over tropical land is improved over the ECHAM default Tiedtke-Nordeng convection parameterization in terms of phase but with a too small afternoon peak. To get a reasonable climate in clouds and energy balance the CCFM scheme needed to be tuned in terms of sub-cloud properties rather strongly (parcel perturbation dT=2.8K, parcel initiation at 350m).

There are two overarching issues with the text.

1. Two base models ECHAM/ECHAM-HAM confusing

   The convection parameterization Tiedtke-Nordeng and CCFM are in the manuscript compared based on sometimes ECHAM-HAM and standard ECHAM. ECHAM-HAM add a two-moment modal aerosol scheme. But the big relevant difference is the microphysics scheme (two-moment and one-moment respectively). That explains rather different behaviour of clouds and radiative fluxes and as a result even the diurnal cycle. I suggest the authors to decide of one base model to show in the main manuscript and move the other plots to the supplement or an appendix. Maybe with the aim at aerosol/convection interactions the ECHAM-HAM should be the primary choice.

2. Explanation of results

   In section 4 several interesting results are presented but such as the sensitivity to the sub-cloud parameter choices, the two convection parameterizations and the "HAM" model component. Explanations are often missing. I do expect from a model developer paper at least an idea why a diurnal cycle changes or clouds are shifting in magnitude and location. I will note a few examples below, but this effort is really important to advance the understanding of parameterizations.

In general, this paper is scientifically interesting. Given a more thorough discussion of results and work on the presentation as outlined in this review this paper can add to the understanding of convective parameterization and I can recommend publication in ACP.

**Specific comments**

3. Line 57, Introduction: "most paramterizations of this type prescribe the cloud spectrum empirically"

   Here you refer to parameterization of the type AS74 as mentioned a few lines above. AS74 though uses a kernel for the interaction of cloud types within the cloud spectrum. They are therefore "dynamic" and not "empirical". Please find a better formulation.

4. L137, Section 2.2.2: "parcel of air from a configurable level"

   This is the paragraph where you describe the tuning setup for the sub-cloud parcel. You need to add the Table 1 and the text explaing the initiation level from L226 in section 6 "Method".

5. L138, Section 2.2.2: "2.8K"

   This value that gives the best results is a rather big value. Typical temperature perturbations used in conveciton schemes are around 1K. Therefore you need to refer to a comparison to other schemes - for example the Tiedtke/Nordeng value used in ECHAM. And then later when discussing Figure 6 you need to explain why such a large value is necessary phyiscally.

6. L146, Section 2.2.2: "initial parcel radii"

   When you say "initial", does that mean that the parcel radii are allowed to change with height? If not, remove "initial". If yes, describe how.

7. L198, Section 2.2.3: "1000 steps is reached" (replace by "are")

   Here and in the conclusion you need to mention the speed of the model runs when comparing Tiedtke/Nordeng with CCFM. How much slower does CCFM run? Is there a more efficient technique?

8. L261, section 4.1: "les liquid and ice when CCFM is used"

   Delete "and ice". Same in L263. Figure does not support that statement for ice.

9. L272-274, section 4.1 and Figure 2:

   Here you need to mention quantitatively that ECHAM-HAM has 3x less IWP that ECHAM. And please try to explain this drastic phenomena beyond the qualitative speculation that the different cloud schemes are responsible.

   You also should explain why there is more LWP sensitivity in ECHAM-HAM due to convection scheme. When looking at fig 2 and 3, it is interesting to note that liquid water in CCFM is significantly located at the lowest model level, while in Tiedtke-Nordeng much is above the boundary layer. This needs to be mentioned in the text and explained.

   This phenomena might be related to the description of shallow convection. Please describe the shallow convection used in CCFM (or lack of as in AS74).

10. L282, section 4.2.1 "CCRM show a negative cloudy bias ... Tiedtke-Nordeng shows a clear positive bias ..." (add "negative" and "positive" for clarity)
    Please explain this.

11. L291, section 4.2.1 "too little cloud .. due to suppression by low-level inversions"

    This explanation is rather unclear. Low-level inversions help stratocumulus.

    One speculation would be that the higher parcel initiation (L-2, L-3) favour deeper updrafts and therefore less shallow convection. Less shallow convection then leaves more moisture in the sub-cloud layer with allow more stratocumulus to be formed (more low cloud).

    An analogous argument can be made with dT in figure 6. High temperature perturbation allow deeper updrafts ... .

12. Figure 8

    Mention the difficulty of CCFM in CRE and explain. Too much low cloud?

13. Figure 12a

    There are two modes in cloud bse updraught velocity. Please explain. Does that represent shallow and deep convection?

**Technical corrections**

14. L146, Section 2.2.2

    "model is run for a range of"

    replace by

    "model is run for a number of"

15. L151, Section 2.2.2

    "cloud-base radii from r1 to max(r,max; z,PBL) where"

    replace by

    "cloud-base radii from r1 to r,max(z,PBL) where"

16. L187, Section 2.2.3

    "where the coefficients"

    replace by

    "where the coefficients are"

17. L359-360, Section 4.2.3

    "As noted aboev, Tiedtke-Nordeng also ... configuration."

    This sentence can be deleted as it has been mentioned already above.

18. L368, Section 4.2.3

    "cloud-base radius and updraught velocity"

    replace by

    "cloud-base radius and the updraught velocity"

19. Figure 2

    All LWP figures should have same color scale for easier comparison.

20. Figure 3

    All LWC figures should have same color scale for easier comparison.

---

## Referee Comment (RC2) · Anonymous Referee #1 · 2 Sep 2016

The paper provides an overview of the behavior of a parameterization for deep convection in the ECHAM6-HAM2 general circulation model. The parameterization is among few for deep convection which include vertical velocities at convective scale, essential for incorporating cloud-aerosol interactions (and more realistic microphysics and cloud radiative interactions generally) in these clouds, as vertical velocity is a major control on activation of liquid droplets and ice crystals. Although this paper does not deal with these interactions, it is important to establish a baseline for such work by demonstrating successful simulations using this parameterization approach. This paper does so successfully.

In my earlier referee report for the initial manuscript evaluation, I detailed a number of suggested revisions. Per *ACPD* procedures, I should have deferred these to the interactive discussion, so aplogies to the editors here. The authors have either responded satisfactorily or incorporated these revisions in the discussion paper as it currently stands. I recommend publication.
* * *

---

## Author Comment (AC1) · 6 Nov 2016

We are grateful to the two anonymous referees for their time and constructive comments on our original manuscript and during the public discussion. All points raised by reviewer #1 were addressed during the access review. We have made a number of alterations in a revised manuscript to address the further points raised by reviewer #2 during the discussion phase, and we hope that the manuscript is now clearer as a result. Responses to individual points, and details of changes to the manuscript, are given below.

**Response to reviewer #2**

*1. Two base models ECHAM/ECHAM-HAM confusing*
*The convection parameterization Tiedtke-Nordeng and CCFM are in the manuscript compared based on sometimes ECHAM-HAM and standard ECHAM. ECHAM-HAM add a two-moment modal aerosol scheme. But the big relevant difference is the microphysics scheme (two-moment and one-moment respectively). That explains rather different behaviour of clouds and radiative fluxes and as a result even the diurnal cycle. I suggest the authors to decide of one base model to show in the main manuscript and move the other plots to the supplement or an appendix. Maybe with the aim at aerosol/convection interactions the ECHAM-HAM should be the primary choice.*

We agree that focussing on ECHAM–HAM makes the manuscript clearer, and have followed this suggestion, moving the standard ECHAM results into the supplement.

*2. Explanation of results*
*In section 4 several interesting results are presented but such as the sensitivity to the sub-cloud parameter choices, the two convection parameterizations and the "HAM" model component. Explanations are often missing. I do expect from a model developer paper at least an idea why a diurnal cycle changes or clouds are shifting in magnitude and location. I will note a few examples below, but this effort is really important to advance the understanding of parameterizations.*

We agree that the explanations of some of the results could be expanded upon. See subsequent points for specific changes made in this regard.

*3. Line 57, Introduction: "most paramterizations of this type prescribe the cloud spectrum empirically"*
*Here you refer to parameterization of the type AS74 as mentioned a few lines above. AS74 though uses a kernel for the interaction of cloud types within the cloud spectrum. They are therefore "dynamic" and not "empirical". Please find a better formulation.*

We agree that this was unclear, and mischaracterised the original AS74 scheme. We have revised the text as follows:

. . . typically defined by their fractional entrainment rates. **In the original derivation, the interaction kernel between cloud types is calculated dynamically based on the bulk dynamic and thermodynamic behaviour of the cloud ensemble; simpler implementations may prescribe the cloud spectrum empirically.** The Convective Cloud Field Model (CCFM; Nober and Graf, 2005; Wagner and Graf, 2010) **couples the dynamical system approach to the cloud spectrum** with an explicit entraining plume model **with embedded microphysics** for each cloud type to predict the spectrum based on the competitive interactions between different cloud types. This provides a promising setup in which to investigate the effects of convective microphysics at the global scale.

*4. L137, Section 2.2.2: "parcel of air from a configurable level"*
*This is the paragraph where you describe the tuning setup for the sub-cloud parcel. You need to add the Table 1 and the text explaing the initiation level from L226 in section 6 "Method".*

We prefer to retain the distinction between the general model description in Section 2 and the specific configuration values chosen for the sensitivity experiments in Section 3. However we have added the sentence:

**Sensitivity to the starting level of the parcel and its buoyancy perturbation will be discussed later.**

*5. L138, Section 2.2.2: "2.8K"*
*This value that gives the best results is a rather big value. Typical temperature perturbations used in conveciton schemes are around 1K. Therefore you need to refer to a comparison to other schemes – for example the Tiedtke/Nordeng value used in ECHAM. And then later when discussing Figure 6 you need to explain why such a large value is necessary phyiscally.*

The likely magnitude of localised temperature perturbations will be very regime-dependent. In particular, values over the ocean (or other uniform surface types) are likely to be small, while those over orographic features and surface-type discontinuities may be significantly larger. Thus we would argue that any choice of a globally-fixed value for this purpose is somewhat arbitrary, which is why in the following paragraph we suggest a future version of the scheme is likely to take regional features into account in choosing the perturbation.

While the choice of perturbation is somewhat arbitrary, it is also tightly coupled with another parameter which is subject to arbitrary tuning in most parameterisations – the entrainment rate. In this work, we stick to the traditional $C_\mu = 0.2$ in Eq. (1), however further experiments have shown that smaller values of $C_\mu$ require smaller values of the temperature perturbation to achieve radiative balance, while improving aspects of the cloud spectrum itself.

The following text has been added at the end of the subsequent paragraph:

**The value of** $2.8$ **K is rather larger than the maximum** $1$ **K used for triggering in Tiedtke–Nordeng, but it is worth noting that the required**

**perturbation in CCFM is strongly correlated with $C_\mu$ and therefore this process is not dissimilar to the common practice of using the Tiedtke–Nordeng entrainment rates for tuning ECHAM (as in e.g. Mauritsen et al., 2012) rather than setting them based on physical considerations. The variation of $C_\mu$ is discussed further in Labbouz et al. (2016).**

and the following in the second paragraph of Section 4.2.1 where Figure 6 is introduced:

**That such a large perturbation is required may be an indication that the customary entrainment parameter $C_\mu = 0.2$ as used in Wagner and Graf (2010) is too large for the convective regimes involved, since smaller values do allow radiative balance to be achieved with a weaker perturbation (not shown).**

*6. L146, Section 2.2.2: "initial parcel radii"*
*When you say "initial", does that mean that the parcel radii are allowed to change with height? If not, remove "initial". If yes, describe how.*

Yes, parcel radii will change with height to maintain mass continuity during acceleration/deceleration and entrainment/detrainment. This is a standard part of the entraining plume model formulation as given in the references, and alluded to in Section 2.2.1: "...determine the evolution of...$r$ from cloud base to cloud top".

*7. L198, Section 2.2.3: "1000 steps is reached" (replace by "are")*
*Here and in the conclusion you need to mention the speed of the model runs when comparing Tiedtke/Nordeng with CCFM. How much slower does CCFM run? Is there a more efficient technique?*

"is" is correct: the subject is "a limit of. . ." (singular), not "1000 steps".

These particular limits are of little relevance for model speed as they are only invoked in rare instances, and most of the computational cost is in the entraining plume models rather than the iterative solution of the Lotka–Volterra equations. As a complex research parameterisation, CCFM is of course considerably slower than a well-established and optimised bulk scheme. There is undoubtedly significant scope for improving its computational efficiency, but we feel this is outside the scope of the present work, whose focus is on evaluating the output of the model.

> *8. L261, section 4.1: "les liquid and ice when CCFM is used"*
> *Delete "and ice". Same in L263. Figure does not support that statement for ice.*

We agree, and have removed the reference to ice here.

> *9. L272-274, section 4.1 and Figure 2:*
> *Here you need to mention quantitatively that ECHAM-HAM has 3x less IWP that ECHAM. And please try to explain this drastic phenomena beyond the qualitative speculation that the different cloud schemes are responsible. You also should explain why there is more LWP sensitivity in ECHAM-HAM due to convection scheme. When looking at fig 2 and 3, it is interesting to note that liquid water in CCFM is significantly located at the lowest model level, while in Tiedtke-Nordeng much is above the boundary layer. This needs to be mentioned in the text and explained. This phenomena might be related to the description of shallow convection. Please describe the shallow convection used in CCFM (or lack of as in AS74).*

A detailed investigation of the reasons why (these versions of) ECHAM–HAM and ECHAM differ in their representation of ice cloud is outside the scope of this paper.

However, due to the weak observational constraints available, IWP remains highly dependent on the model tuning state which is usually determined based on better-constrained quantities (see e.g. Lohmann and Ferrachat, 2010; Mauritsen et al., 2012).

There is no separate shallow convection scheme used; it is assumed that the smallest, most-rapidly-entraining clouds represent shallow cumulus. The following text has been added in Section 2.2 to clarify this:

> **There is no separate shallow convection scheme, with CCFM aiming to represent both shallow and deep cloud. The smallest clouds have higher entrainment rates and hence grow less, while larger clouds are more likely to produce deep convection.**

The following text has been added in Section 4.2.1 to address the low-level liquid water cloud in CCFM:

> **CCFM also shows a concentration of liquid water in the lowest model levels, separated from that in the free troposphere by a drier layer. This may be related to the entraining plume framework being more suited to deep than shallow convection, or to differences between CCFM and Tiedtke–Nordeng in the coupling with the turbulent mixing in the boundary layer scheme.**

> *10. L282, section 4.2.1 "CCRM show a negative cloudy bias ... Tiedtke-Nordeng shows a clear positive bias ..." (add "negative" and "positive" for clarity)*
> *Please explain this.*

The text has been changed to clarify the sense of the biases, although note that these are in the other direction to that suggested in the comment, as per the words *cloudy*

and *clear*: CCFM's *cloudy* bias is a positive cloud cover bias, while Tiedke–Nordeng's *clear* bias is a negative cloud cover bias.

> . . . CCFM shows a **positive cloud cover bias (i.e. too cloudy)** over the western side of the ocean basins, while Tiedtke–Nordeng shows a **negative bias (i.e. too clear)** over the eastern side.

> *11. L291, section 4.2.1 "too little cloud .. due to suppression by low-level inversions"*
> *This explanation is rather unclear. Low-level inversions help stratocumulus. One speculation would be that the higher parcel initiation (L-2, L-3) favour deeper updrafts and therefore less shallow convection. Less shallow convection then leaves more moisture in the sub-cloud layer with allow more stratocumulus to be formed (more low cloud). An analogous argument can be made with dT in figure 6. High temperature perturbation allow deeper updrafts ... .*

We agree that this statement is somewhat unclear. It is of course true that inversions at a low level, but above the lifting condensation level (LCL), are key to the formation of stratocumulus. However, an inversion *below* the LCL will trap moisture in the surface layer (consistent with the behaviour noted in point 9) rather than allowing it to be lifted to form stratocumulus. In global models, it is commonly lifting by the shallow convection scheme, rather than turbulent vertical mixing by the boundary layer scheme, which forms much of the condensate in stratocumulus regions (see e.g. Morcrette and Petch, 2010).

Looking at the CCFM cloud-top distributions in these regions, there is virtually no deep convection for any parcel initiation level, and we would rather expect any deep convection to remove moisture from the boundary layer, thus diminishing rather than enhancing the stratocumulus deck.

We have modified the text to make this clearer:

> . . . perhaps due to suppression by **near-surface inversions below the LCL**. **It should be noted in this context that in the absence of a specific stratocumulus parameterisation, in global models it is often detrainment from the convection scheme which produces much of the condensate in stratocumulus regions – this can be seen for example in Figure 6a of Morcrette and Petch (2010).**

> *12. Figure 8*
> *Mention the difficulty of CCFM in CRE and explain. Too much low cloud?*

We have added the following text where this figure is introduced:

> **This does result in an increased RMSE in the net CRE when using CCFM.**

We have also changed the text later in this paragraph to further discuss the reasons for the difference in CRE:

> These are aspects that are very sensitive to **the vertical position of clouds, which controls the balance between their SW and LW effects; this is strongly influenced both by the** tuning of the large-scale cloud scheme **and convective entrainment.** It is likely that **a reduction of** $C_\mu$ **(as mentioned previously and discussed further in Labbouz et al., 2016)** would yield an improvement here **through a reduction of low cloud, as would** re-tuning without the constraint that both Tiedtke–Nordeng and CCFM should be in balance with the same parameter values**.**

*13. Figure 12a*
*There are two modes in cloud bse updraught velocity. Please explain. Does*
*that represent shallow and deep convection?*

The aim in this paper is not to discuss these new outputs in detail, which we expect
to explore further in subsequent work, but rather to evaluate the performance in the
global model. However, these two modes do indeed broadly correspond to shallower
and deeper cloud regimes, yes. The following text has been added:

**The bimodality broadly corresponds to shallower and deeper cloud**
**regimes (with stronger updraughts at the base of the latter), although**
**there remains considerable variation within each class (not shown).**

*14. L146, Section 2.2.2*
*"model is run for a range of"*
*replace by*
*"model is run for a number of"*

We've deleted "a range of", which makes the sentence clearer and more succinct.

*15. L151, Section 2.2.2*
*"cloud-base radii from r1 to max(r,max; z,PBL) where"*
*replace by*
*"cloud-base radii from r1 to r,max(z,PBL) where"*

The expression as in the manuscript is the correct version.

*16. L187, Section 2.2.3*

[Figure]

*"where the coefficients"*
*replace by*
*"where the coefficients are"*

We have inserted "are given by" here.

*17. L359-360, Section 4.2.3*
*"As noted aboev, Tiedtke-Nordeng also ... configuration."*
*This sentence can be deleted as it has been mentioned already above.*

Deleted.

*18. L368, Section 4.2.3*
*"cloud-base radius and updraught velocity"*
*replace by*
*"cloud-base radius and the updraught velocity"*

We have left the text intact, because inserting "the" would suggest that "updraught velocity" is no longer referring to cloud base.

*19. Figure 2*
*All LWP figures should have same color scale for easier comparison.*
*20. Figure 3*
*All LWC figures should have same color scale for easier comparison.*

These points are largely moot now that the non-HAM panels have been moved into the supplement. However, while we use the same scales for Tiedtke–Nordeng and CCFM, we prefer to use different scales for ECHAM and ECHAM–HAM since the aim is to clearly show the difference between convection schemes in each case, rather than the (larger) difference between the two base models.

**Other changes**

We have also updated the first author's affiliation with:

∗ **now at the European Centre for Medium-Range Weather Forecasts, Reading, UK**

**References**

Labbouz, L., Kipling, Z., Stier, P., and Protat, A.: How well can we represent the spectrum of convective clouds in a climate model?, J. Atmos. Sci., submitted, 2016.

Lohmann, U. and Ferrachat, S.: Impact of parametric uncertainties on the present-day climate and on the anthropogenic aerosol effect, Atmos. Chem. Phys., 10, 11 373–11 383, 2010.

Mauritsen, T., Stevens, B., Roeckner, E., Crueger, T., Esch, M., Giorgetta, M., Haak, H., Jung-claus, J., Klocke, D., Matei, D., Mikolajewicz, U., Notz, D., Pincus, R., Schmidt, H., and Tomassini, L.: Tuning the climate of a global model, Journal of Advances in Modeling Earth Systems, 4, M00A01, doi:10.1029/2012MS000154, 2012.

Morcrette, C. J. and Petch, J. C.: Analysis of prognostic cloud scheme increments in a climate model, Q. J. R. Meteorol. Soc., 136, 2061–2073, doi:10.1002/qj.720, 2010.

Nober, F. J. and Graf, H. F.: A new convective cloud field model based on principles of self-organisation, Atmos. Chem. Phys., 5, 2749–2759, doi:10.5194/acp-5-2749-2005, 2005.

Wagner, T. M. and Graf, H.-F.: An Ensemble Cumulus Convection Parameterization with Explicit Cloud Treatment, Journal of the Atmospheric Sciences, 67, 3854–3869, doi:10.1175/2010JAS3485.1, 2010.